# Outcomes of different steroid dosing regimens in critical Covid-19 pneumonia at a Kenyan hospital: A retrospective cohort study

John Otieno Odhiambo[1]*, Jasmit Shah[2], Nancy Kunyiha[1], Charles Makasa[1], Felix Riunga[1]

1 Healthcare Practitioner at the Department of Internal Medicine, Aga University, Nairobi, Kenya,
2 Statistician at the Department of Internal Medicine and Brain and Mind Institute, Aga Khan University, Nairobi, Kenya

* john.odhiambo2@scholar.aku.edu

## Abstract

### Background

Among therapeutic options for severe and critical COVID- 19 infection, dexamethasone six milligrams once daily for ten days has demonstrated mortality benefit and is guideline recommended at this dose. In practice, variable doses of steroids have been used, especially in critical care settings. Our study aimed to determine the pattern of steroid dosing and outcomes in terms of critical care mortality, occurrence of dysglycaemias, and occurrence of superadded infections in patients with critical COVID-19.

### Methods

A retrospective cohort study was carried out on all eligible patients admitted to the Aga Khan University Hospital, Nairobi, with critical COVID-19 between 1st March 2020 and 31st December 2021. The intervention of interest was corticosteroids quantified as the average daily dose in milligrams of dexamethasone. A steroid dose of six milligrams once a day was compared to high dose steroid dosing, which was defined as any dose greater than this. The primary outcome measure was ICU mortality and secondary outcomes included occurrence of dysglycaemias, superadded infections and duration of critical care admission.

### Results

The study included 288 patients. The median age was 61.2 years (IQR: 49.7, 72.5), with 71.2% of patients being male. The most common comorbidities were diabetes mellitus (60.7%), hypertension (58%), and heart disease (12.2%). The average oxygen saturation and C-reactive protein at admission were 82% [IQR: 70.0–89.0]and 113.0 [IQR: 54.0–186.0], respectively. Fifty-eight percent of patients received a standard dose (6mg) of steroids. The mortality rate was higher in the high-dose group compared to the standard-dose group; however, the difference was not statistically significant (47.9% vs 43.7% p = 0.549). The two most common steroid associated adverse effects were uncomplicated hyperglycemia (62.2%) and superimposed bacterial pneumonia (20.1%). The high-dose group had a

**Data Availability Statement:** All relevant data are within the manuscript and its Supporting Information files.

**Funding:** The author(s) received no specific funding for this work.

**Competing interests:** The authors have declared that no competing interests exist.

higher incidence of uncomplicated hyperglycemia compared to the standard-dose group (63.6% vs 61.1%). However, the incidence of diabetic ketoacidosis was lower in the high dose group (0.6% vs 6.6%). Oxygen saturation at admission was associated with survival where it was lower among non-survivor patients with critical COVID-19.

## Conclusion

The study found that high-dose steroids in the treatment of critically ill patients with COVID-19 pneumonia did not confer any mortality benefit and were associated with an increased risk of dysglycemia and superimposed infections.

## Definition of terms

### COVID-19 pneumonia

Clinical and radiologic features consistent with pneumonia and a positive SARS CoV-2 PCR from a respiratory specimen.

### Severe COVID-19

Those who are confirmed to have COVID-19 with oxygen saturations less than 94%, requiring oxygen supplementation.

### Critical COVID-19

Those who are confirmed to have COVID-19 and have respiratory failure, septic shock, and/or multiple dysfunction. $\geq 0.6$mmol/L.

### High dose steroid

Dexamethasone dose higher than 6mg Once daily or its equivalence.

### Low dose steroid/standard steroid dose

Dexamethasone 6mg once daily or its equivalence.

### Dysglycemia

Hyperglycemia, diabetic ketoacidosis or hyperglycemic hyperosmolar state.

### Diabetic ketoacidosis

Anyone confirmed to have Diabetes Mellitus with normal or high serum glucose level, blood pH less than 7.3/serum bicarbonate level <18.0mmol/L and serum ketone level.

### Hyperglycemic hyperosmolar state

Plasma glucose level >33mmol/L and increased effective plasma osmolality >320mOsm/Kg in the absence of ketoacidosis.

### Bloodstream infection

Isolation of a pathogenic microbe from at least one blood culture or isolation of an organism considered to be a skin commensal from at least two sets of blood cultures in the presence of a compatible syndrome as ascertained by an infectious diseases specialist in the study team.

### Urinary tract infection

Defined as the growth of a bacterium or fungus in a cultured urine sample from a patient with clinical symptoms and/or the consideration of such urinary infection as clinically significant by an infectious disease specialist in the study team.

### Bacterial/viral/mycobacterial/fungal pneumonia

Growth of a bacterial or fungal isolate from a respiratory sample (sputum/tracheal aspirate/broncho alveolar lavage fluid) or a positive PCR from a respiratory sample with a compatible clinical syndrome as ascertained by an infectious disease specialist in the study team.

### Heart disease

Ischemic heart disease, heart failure, or arrhythmias.

## Introduction

Steroid use as a cornerstone of the management of severe to critical COVID-19 disease has been established by multiple randomized controlled trials and meta-analysis [1–3]. Dexamethasone was the first steroid to be studied and was found to lower 28-day mortality among patients requiring mechanical ventilation or oxygen supplementation when given at a dose of 6 milligrams once daily for 10 days or until discharge [1].

Initial data on what the optimal dose of steroids should be was scant, and as a result, higher dose steroids were used in several centres [4–7]. More recently, large, randomized controlled trials have suggested that high dose steroids are no more effective than standard dose steroids and may even be associated with mortality and adverse effects [8, 9]. Uncertainty remains on whether high dose steroids would benefit those requiring high oxygen demands, including critically ill patients [10]. Of note is that very little data is available from Africa on treatment of severe and critical COVID-19 with different steroid regimens and outcomes [11]. In the ACCCOS study which investigated outcomes of patients admitted with COVID-19 in intensive or high care units across 10 African countries, steroid therapy was associated with lower risk of mortality [12]. However, no data was given on the dose, duration or complications associated with steroids. In the RECOVERY trial investigating 12 mg vs 6 mg of dexamethasone, only 15 patients out of 1272 randomised patients were from an African country [9]. Although steroids have improved clinical outcomes, their use, particularly at higher doses, has raised concerns for adverse events including superimposed infections and dysglycaemias [9, 13].

We therefore conducted a retrospective cohort study among patients admitted with critical COVID-19 at a referral hospital to investigate the pattern of steroid use in the management of these patients, and the effect on mortality, dysglycaemias and superadded infections.

## Materials and methods

### Study design

This was a retrospective cohort study conducted between 1st March 2020 and 31st December 2021.

## Setting

The study was conducted at the Aga Khan University Hospital, Nairobi (AKUH, N).

## Population

We included patients admitted with critical COVID-19 pneumonia between 1st March 2020 and 31st December 2021. Critical COVID-19 pneumonia with defined as those with COVID-19 pneumonia with respiratory failure, septic shock, or multiple organ dysfunction. Those who were pregnant and those who were transferred from other facilities to the critical care units were excluded from our study. Those with missing data that would make the objectives of the study un-assessable were also excluded.

## Data collection

Patients were identified from a database maintained by the critical care team that included all patients admitted to the hospital between 1st March 2020, and 31st December 2021. Each file was identified by a unique number allocated to every patient on admission to the facility, and the file was retrieved physically from the hospital's medical records department. The data collected were entered into REDCap [14]. The data collection was done from October 2022 to January 2023 and all data was de-identified for analysis. The variables of interest included demographic information, date of COVID-19 diagnosis, comorbid conditions, the average daily dose of steroid given in dexamethasone equivalents expressed in milligrams, duration of mechanical ventilation and critical care stay, outcome (discharged alive or dead), and adverse effects associated with corticosteroid use. The intervention of interest was the use of corticosteroids, and the primary outcome was the effect of different steroid doses (standard dose of six milligrams once a day of dexamethasone versus high dose steroids, which was defined as any dose greater than this) on ICU mortality. The secondary outcomes included adverse effects linked to steroid use, (specifically dysglycemia and infections), duration of critical care admission and duration of ventilatory support. Since this was a retrospective study, no consent was required and a waiver was sought and obtained from the Aga Khan University Institutional Scientific and Ethics Review Committee (2022/ISERC-32(v1)) and National Commission for Science, Technology and Innovation (NACOSTI) (Ref 494615).

## Data analysis

The data collected was entered into REDCap database and then exported to IBM Statistical Package for the Social Sciences version 22.00(SPSS) for analysis. Categorical data such as comorbid conditions, dysglycemia and infections were presented as frequencies and percentages whereas continuous data for instance age of the patients, durations of ventilatory support and duration of critical care stay were presented as means and standard deviations. Univariate analysis was performed using Chi-squared or Fishers Exact test for categorical data whereas students t-test or Kruskal Wallis Test for continuous data when comparing more than 2 groups. Group comparisons were done based on standard dose (Dexamethasone 10mg OD/ equivalence for 10 days) versus high dose (more than dexamethasone 10mg OD/equivalence for 10 days).

## Results

### Characteristics of the study population

Four hundred and sixty-eight patients were admitted to the critical care units between 1st March 2020 and 31st December 2021. Out of this, 293 were on ventilator support. Five patients

**Table 1. Demographic and clinical characteristics of 288 patients.**

| Age (years) | | 61.2 [49.7, 72.5] | |
|---|---|---|---|
| Gender | Male | 205 | 71.2% |
| | Female | 83 | 28.8% |
| Comorbidities | Hypertension | 167 | 58.0% |
| | Diabetes | 175 | 60.8% |
| | CKD | 30 | 10.4% |
| | Heart Disease | 35 | 12.2% |
| | Malignancy | 4 | 1.4% |
| | Chronic Lung Disease | 26 | 9.0% |
| | Other | 120 | 41.7% |
| Outcome | Discharge from critical care | 157 | 54.5% |
| | Dead | 131 | 45.5% |
| Dysglycemia | Uncomplicated Hyperglycemia | 179 | 62.2% |
| | Diabetic Ketoacidosis | 12 | 4.2% |
| | Hyperglycemic Hyperosmolar state | 2 | 0.7% |
| | None | 95 | 33.0% |
| Infections | Urinary Tract Infections | 27 | 9.4% |
| | Superimposed Pneumonia | 58 | 20.1% |
| | Bloodstream Infection | 48 | 16.7% |
| | None | 194 | 67.4% |
| Steroid Dose Category | Standard (6mg) | 167 | 58.0% |
| | High (>6mg) | 121 | 42.0% |
| Duration of steroid use (Days) | | 10.0 [9.0, 15.0] | |
| Duration of critical care admission–all patients (Days) | | 11.0 [6.0, 18.0] | |
| Duration of critical care admission- non-survivors (Days) | | 10.0 [5.0, 17.0] | |
| SP02 at presentation | | 82.0 [70.0, 89.0] | |
| CRP at admission | | 113.0 [54.0, 186.0] | |

*Continuous variables presented as median [IQR]

were excluded because of incomplete records. Therefore, 288 patients were included in the study. The median age of the patients was 61.2 years (IQR: 49.7–72.5), with 71.2% being male. The most common comorbidities were diabetes mellitus, hypertension, and heart disease reported in 60.7%, 58%, and 12.2% of the patients, respectively. The median oxygen saturation and C-reactive protein at admission were 82.0% (IQR: 70.0–89.0) and 113.0 (IQR:54.0–186.0), respectively. The median duration to discharge from critical care unit was 11 days (IQR: 6–18) and the mortality rate was 45.5%. The median age of non survivors was higher than that of survivors (70.0 years (IQR:54.8–77.7) versus 57.7 years (IQR:47.9–66.7); p<0.001)). Table 1 details the study population characteristics.

## Patterns of steroid use

Fifty eight percent of patients received a standard dose of steroids, while 42% received high dose steroids. The median dose of the high dose group 14.0 mg (IQR:9.5–18.3). The average daily dose of steroid in the high dose group was 15mg. The median duration of steroid use was 10 days (IQR:9–15). When comparing characteristics between those that received standard-dose steroids versus those that received high-dose steroids, CKD, dysglycemia, duration of steroid use and duration of critical care admission were statistically significant. Duration of

**Table 2. Comparison between standard and high dose steroid groups (n = 288).**

| | | Standard Dose | | High Dose | | P Value |
|---|---|---|---|---|---|---|
| | | (n = 167) | | (n = 121) | | |
| Age (years) | | 60.2 [49.5, 71.4] | | 64.2 [50.2, 73.4] | | 0.362 |
| Gender | Male | 120 | 71.9% | 85 | 70.2% | 0.793 |
| | Female | 47 | 28.1% | 36 | 29.8% | |
| Comorbidities | Hypertension | 99 | 59.3% | 68 | 56.2% | 0.630 |
| | Diabetes | 105 | 62.9% | 70 | 57.9% | 0.395 |
| | CKD | 23 | 13.8% | 7 | 5.8% | 0.032 |
| | Heart Disease | 17 | 10.2% | 18 | 14.9% | 0.274 |
| | Malignancy | 3 | 1.8% | 1 | 0.8% | 0.642 |
| | Chronic Lung Disease | 17 | 10.2% | 9 | 7.4% | 0.533 |
| | Other | 66 | 39.5% | 54 | 44.6% | 0.399 |
| Dysglycemia | Uncomplicated Hyperglycemia | 102 | 61.1% | 77 | 63.6% | 0.025 |
| | Diabetic Ketoacidosis | 11 | 6.6% | 1 | 0.8% | |
| | Hyperglycemic Hyperosmolar | 0 | 0.0% | 2 | 1.7% | |
| | None | 54 | 32.3% | 41 | 33.9% | |
| Infections | Urinary Tract Infections | 12 | 7.2% | 15 | 12.4% | 0.154 |
| | Superimposed Pneumonia | 27 | 16.2% | 31 | 25.6% | 0.054 |
| | Bloodstream Infection | 25 | 15.0% | 23 | 19.0% | 0.424 |
| Duration of steroid use (Days) | | 10.0 [7.0, 12.0] | | 14.0 [10.0, 19.0] | | <0.001 |
| Outcome | Discharge from critical care | 94 | 56.3% | 63 | 52.1% | 0.549 |
| | Dead | 73 | 43.7% | 58 | 47.9% | |
| Duration of critical care admission- all patients (Days) | | 11.0 [6.0, 16.0] | | 11.0 [8.0, 23.0] | | 0.132 |
| Duration of critical care admission-non survivors (Days) | | 8.0 [4.0, 14.0] | | 11.5 [6.0, 17.0] | | 0.036 |
| SPO2 at presentation | | 82.0 [70.0, 89.0] | | 82.0 [70.0, 89.0] | | 0.560 |
| CRP at admission | | 125.0 [62.5, 199.5] | | 91.0 [45.0, 161.0] | | 0.020 |

*Continuous variable presented as median [IQR]

steroid use was longer in the high dose group compared to the standard dose group (14 days (IQR: 10–19) versus 10 days (IQR: 7–12), p<0.001). Uncomplicated hyperglycemia was higher in the high-dose group compared to the standard-dose group (63.6% vs 61.1%, p = 0.025). Duration of critical care admission in non-survivors was longer in the high-dose group compared to the standard-dose group (11.5 days (IQR: 6–17) vs 8 days (IQR: 4–14), p = 0.036). CRP was found to be statistically significant among the dose groups where the CRP levels were low in the high dose group as compared to the standard dose group; 91.0 (IQR: 45.0–161.0) versus 125.0 (IQR: 62.5–199.5); p = 0.020. Details of the comparison between steroid dose groups are presented in Table 2.

## Outcomes

The overall ICU mortality was 45.5%. The median duration of critical care admission in non-survivors was 10 days (IQR: 5–17). When comparing the characteristics between survivors and non-survivors, age, heart disease, infections, and duration of steroid use were statistically significant. Survivors were younger than non-survivors (57.7 years vs 70.0 years, p<0.001). The median oxygen saturation at admission was lower in the non-survivors' group as compared to the survivors group, 75.0 (IQR: 68.087.5) versus 85.0 (IQR: 75.0–89.5) and the difference was statistically significant (p<0.001). There were more superimposed bacterial pneumonias

**Table 3. Comparison between outcomes of 288 patients (survivors vs non survivors).**

| | | Survivors | | Non-survivors | | P Value |
|---|---|---|---|---|---|---|
| | | (n = 157) | | (n = 131) | | |
| Age (years) | | 57.7 [47.9, 66.7] | | 70.0 [54.8, 77.7] | | <0.001 |
| Gender | Male | 109 | 69.4% | 96 | 73.3% | 0.515 |
| | Female | 48 | 30.6% | 35 | 26.7% | |
| Comorbidities | Hypertension | 85 | 54.1% | 82 | 62.6% | 0.153 |
| | Diabetes | 88 | 56.1% | 87 | 66.4% | 0.090 |
| | CKD | 12 | 7.6% | 18 | 13.7% | 0.121 |
| | Heart Disease | 13 | 8.3% | 11 | 16.8% | 0.031 |
| | Malignancy | 2 | 1.3% | 2 | 1.5% | 1.000 |
| | Chronic Lung Disease | 11 | 7.0% | 15 | 11.5% | 0.218 |
| | Other | 58 | 36.9% | 62 | 47.3% | 0.093 |
| Dysglycemia | Uncomplicated Hyperglycemia | 93 | 59.2% | 86 | 65.6% | 0.106 |
| | Diabetic Ketoacidosis | 4 | 2.5% | 8 | 6.1% | |
| | Hyperglycemic Hyperosmolar | 2 | 1.3% | 0 | 0.0% | |
| | None | 58 | 36.9% | 37 | 28.2% | |
| Infections | Urinary Tract Infections | 14 | 8.9% | 13 | 9.9% | 0.840 |
| | Superimposed Pneumonia | 21 | 13.4% | 37 | 28.2% | 0.002 |
| | Bloodstream Infection | 18 | 11.5% | 30 | 22.9% | 0.011 |
| Steroid Dose Category | Standard (6mg) | 94 | 59.9% | 73 | 55.7% | 0.549 |
| | High (>6mg) | 63 | 40.1% | 58 | 44.3% | |
| Duration of steroid use (Days) | | 11.0 [10.0, 16.0] | | 10.0 [6.0, 14.0] | | 0.005 |
| SPO2 at presentation | | 85.0 [75.0, 89.5] | | 75.0 [68.0, 87.5] | | <0.001 |
| CRP at admission | | 122.5 [56.0, 199.0] | | 107.0 [52.0, 179.0] | | 0.260 |

*Continuous variable presented as median [IQR]

among non-survivors compared to the survivors (28.2% vs 13.4%, p = 0.002). Details of the comparison between outcomes are presented in Table 3.

## Discussion

In this single centre study, use of higher doses of corticosteroids in the management of critical COVID-19 pneumonia did not confer any mortality benefit and was associated with significantly more dysglycaemia as well as more than a two-fold increase in the occurrence of superimposed pneumonia. In the RECOVERY steroid dose study by Abbas [9], there was an increase in mortality, as well as an excess of superadded pneumonias and dysglycaemias, although the study population comprised of patients requiring simple oxygen supplementation and did not include critical COVID-19 pneumonia. In the COVID STEROID 2 trial by Munch et al, patients requiring high levels of oxygen supplementation, including invasive and non-invasive mechanical ventilation were included [8]. There was no significant mortality benefit for high dose compared to low dose steroids in this trial. There was no difference in superadded infections noted [8]. A meta-analysis by Pitre and colleagues suggested that higher doses of steroids probably reduced mortality with an uncertain effect on nosocomial infections [10]. Notably, the meta-analysis did not include data from the RECOVERY steroid dose study. In summary, our study is in line with most data, suggesting no added benefits and possible harm of high dose steroids in patients with critical COVID 19 pneumonia.

Our patient population was slightly older (61.2 years versus 56 years), with more males (71.2% versus 60.6%) and with diabetes more prevalent (60.8% versus 38%) than that seen in the ACCCOS cohort, which characterized patients with COVID-19 pneumonia admitted to critical care units in Africa [12]. The overall mortality rate was 45.5% which was similar to that seen in the ACCCOS cohort (48.2%) [12]. Non survivors were more likely to be older and have an underlying comorbid condition compared to survivors. The high prevalence of diabetes mellitus and the increase in dysglycaemias seen in the high steroid dose group present more evidence to use standard dose steroids, even in patients critically ill with COVID-19. Though not studied in our population, diabetes, and steroid use, with environmental factors, seemed to be the main risk factor for the mucormycosis outbreak seen in patients with COVID-19 pneumonia in India [13, 15].

In COVID-19, organ damage is caused by inflammation, as evidenced by elevated inflammatory markers in critical COVID-19 patients. In our study, the median C-reactive protein (CRP) level at admission was 113, and it was lower in the high-dose group than in the standard-dose group. The decision to administer steroids may have been at the discretion of the treating doctor and influenced by other factors rather than the level of inflammation. Furthermore, the difference in CRP levels between survivors and non-survivors was not statistically significant. However, the non-survivors had significantly lower oxygen saturation at presentation than the survivors. This was confirmed by a retrospective cohort study in Peru, which found that an oxygen saturation below 90% upon admission was a strong predictor of in-hospital mortality for adult COVID-19 patients [16].

Our study had several limitations. Due to its retrospective nature, missing data and uncontrolled biases may have influenced the results. However, only five patients were excluded from the study because of incomplete records. The effect of various COVID waves and variants was not assessed. Additionally, we did not include data on the different adjunct treatments for COVID-19 such as remdesivir, and anticoagulation which may have been major confounders on the outcomes. The study in a single centre may also limit generalizability of the findings. Our study is however, to the best of our knowledge, the first to attempt to investigate outcomes of high versus low dose steroids in critical COVID-19 pneumonia in an African population. Though it is a single centre study, a relatively large number of patients were included.

## Conclusion

In this study, high dose steroids for the management of critical COVID-19 pneumonia did not improve mortality outcomes and were associated with more adverse events, specifically dysglycaemias and superadded infections.

## Supporting information

**S1 Checklist. STROBE statement—checklist of items that should be included in reports of observational studies.**
(DOCX)

**S1 Data.**
(XLSX)

## Acknowledgments

We would like to acknowledge the following people:

1. Mr. Sabbath Kivuva, from the library department, for his contribution towards document formatting.

2. Staff at the Records department, led by Mr. Kaburia, for their help in retrieving the patients' records.

## Author Contributions

**Conceptualization:** John Otieno Odhiambo, Nancy Kunyiha, Felix Riunga.

**Data curation:** Jasmit Shah.

**Formal analysis:** Jasmit Shah.

**Investigation:** John Otieno Odhiambo, Charles Makasa.

**Methodology:** Jasmit Shah.

**Project administration:** John Otieno Odhiambo.

**Resources:** Nancy Kunyiha, Felix Riunga.

**Supervision:** Nancy Kunyiha, Felix Riunga.

**Writing – original draft:** John Otieno Odhiambo.

**Writing – review & editing:** Jasmit Shah, Nancy Kunyiha, Felix Riunga.

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
