## [Decision Letter · Decision Letter 0]

21 Dec 2023

PONE-D-23-24666OUTCOMES OF DIFFERENT STEROID DOSING REGIMENS IN CRITICAL COVID-19 PNEUMONIA: A RETROSPECTIVE COHORT STUDYPLOS ONE

Dear Dr. Odhiambo,

Thank you for submitting your manuscript to PLOS ONE. After careful consideration, we feel that it has merit but does not fully meet PLOS ONE’s publication criteria as it currently stands. Therefore, we invite you to submit a revised version of the manuscript that addresses the points raised during the review process. Please revise.

We look forward to receiving your revised manuscript.

Kind regards,

Academic Editor

PLOS ONE

Journal Requirements:

4. In this instance it seems there may be acceptable restrictions in place that prevent the public sharing of your minimal data. However, in line with our goal of ensuring long-term data availability to all interested researchers, PLOS’ Data Policy states that authors cannot be the sole named individuals responsible for ensuring data access (http://journals.plos.org/plosone/s/data-availability#loc-acceptable-data-sharing-methods).

Reviewers' comments:

Reviewer's Responses to Questions

**Comments to the Author**

1. Is the manuscript technically sound, and do the data support the conclusions?

Reviewer #1: Partly

Reviewer #2: Yes

2. Has the statistical analysis been performed appropriately and rigorously? 

Reviewer #1: No

Reviewer #2: Yes

3. Have the authors made all data underlying the findings in their manuscript fully available?

Reviewer #1: Yes

Reviewer #2: Yes

4. Is the manuscript presented in an intelligible fashion and written in standard English?

Reviewer #1: Yes

Reviewer #2: Yes

5. Review Comments to the Author

Reviewer #1: Odhiambo J.O. and collaborators performed a single center retrospective observational

study to investigate the effect of different posology of corticosteroids (i.e.,

dexamethasone 6 mg vs. higher equivalent dosages) in patients suffering from critical

COVID-19 pneumonia admitted to ICU of the Aga Khan Hospital in Nairobi, Kenya. I

personally worked in a European COVID ICU during the pandemic, and it was very

complicated and stressful. I imagine working in a low and middle-income country ICU in

such a situation could be exhausting; thus, I would like to thank you for conducting this

research effort, and I hope this review may help you.

The primary outcome of the study was mortality. Secondary outcomes were the

occurrence of superinfections, dysglycemia, and duration of ICU length of stay.

Two-hundred-eighty-eight patients entered the analysis. 48% received dexamethasone 6

mg; 52% higher dosages. Overall, 45.5% of patients died. No difference in mortality was

observed in the two groups (43.7% vs. 47.9%, p=0.549). Higher corticosteroid dosages

seemed to be associated with a higher incidence of dysglycemia and superinfection

events.

The authors have provided a description of an extensively studied topic but considered a

population rarely enrolled in trials and studies. Indeed, the major strength of this work is

the completely African population included. This aspect, in my opinion should be underlined,

both in the title, as well as in the abstract.

Nevertheless, the manuscript, in its current form, requires some improvements and

presents a lot of methodological issues and limitations that should be thoroughly

addressed:

Follow STROBE guidelines to report observational study results, as required by

the Journal publication criteria.

Please specify better the primary outcome: did you assess ICU mortality,

hospital mortality, or 28-day mortality? Moreover, I would suggest adding the

primary outcome result to the main text as well as in the table.

Please, it could be interesting and could improve the quality of the manuscript to

report 1) at least a score or parameters of illness severity (i.e., P/F ratio,

Respiratory Rate, SOFA or SAPS score…); 2) the number of patients who

needed mechanical ventilation; 3) the kind of steroid administered; 4) the timing

of corticosteroid therapy starting compared to ICU admission.

Univariate statistic tests used are completely described, but I would suggest

adjusting for possible confounders (i.e., age, sex, comorbidities, the severity of

illness) for a better evaluation of steroid dosage impact on outcome. Moreover,

if data permits, try to perform a survival analysis (and try to construct a Kaplan-

Meier curve figure).

I would suggest reporting outcome measures in the outcome results section, not

in the population characteristic section.

In the limitation section of the discussion, you stated that missing data may have

influenced the results. Please report the number of missing data for each

variable.

I would suggest reporting Table 1, joined with Table 2, and reporting percentage

of subjects in Table 2 columns.

Reviewer #2: The study is good but, being a retrospective study it has its own limitations as has been pointed by the authors.A few questions for the authors-

1) They have mentioned high dose of steroids as any dose more than 6 mg dexamethasone, were patients on any other steroids like prednisone, methylprednisolone?When were the steroids started in these patients, as in within how many days of onset of symptoms?

2) Was there any comparison between the survivors and non survivors in terms of oxygen requirements, ventilators-NIV or invasive?

3) Was there any correlation between the d Dimer levels amongst the survivors and non survivor groups?

4) How many patients were on Remdesevir considering that these were the more severe cases and was there any correlation on this?

5)How many of these patients also received LMWH or oral anticoagulants and did this affect the outcome of the study?

6) How many patients underwent a bronchoscopy and BAL for the diagnosis of pneumonia?

Thank you.

6. PLOS authors have the option to publish the peer review history of their article (what does this mean?). If published, this will include your full peer review and any attached files.

Reviewer #1: **Yes: **Vittorio Scaravilli, MD

Reviewer #2: No

---

## [Author Response · Author response to Decision Letter 0]

5 Feb 2024

Response to Reviewer’s Comments

PLOS ONE

OUTCOMES OF DIFFERENT STEROID DOSING REGIMENS IN CRITICAL COVID-19 PNEUMONIA: A RETROSPECTIVE COHORT STUDY 

Thank you for the comments. Please find below the responses to both reviewers’ comments. 

Reviewer 1 

Reviewer #1: Odhiambo J.O. and collaborators performed a single center retrospective observational study to investigate the effect of different posology of corticosteroids (i.e., dexamethasone 6 mg vs. higher equivalent dosages) in patients suffering from critical COVID-19 pneumonia admitted to ICU of the Aga Khan Hospital in Nairobi, Kenya. I personally worked in a European COVID ICU during the pandemic, and it was very complicated and stressful. I imagine working in a low and middle-income country ICU in such a situation could be exhausting; thus, I would like to thank you for conducting this research effort, and I hope this review may help you. 

The primary outcome of the study was mortality. Secondary outcomes were the occurrence of superinfections, dysglycemia, and duration of ICU length of stay.

Two-hundred-eighty-eight patients entered the analysis. 48% received dexamethasone 6 mg; 52% higher dosages. Overall, 45.5% of patients died. No difference in mortality was observed in the two groups (43.7% vs. 47.9%, p=0.549). Higher corticosteroid dosages seemed to be associated with a higher incidence of dysglycemia and superinfection events.

The authors have provided a description of an extensively studied topic but considered a population rarely enrolled in trials and studies. Indeed, the major strength of this work is the completely African population included. This aspect, in my opinion should be underlined, both in the title, as well as in the abstract.

Response: We have revised and added “At a Kenyan Hospital” in the title.

Nevertheless, the manuscript, in its current form, requires some improvements and presents a lot of methodological issues and limitations that should be thoroughly addressed:

• Follow STROBE guidelines to report observational study results, as required by the Journal publication criteria.

Response: Thank you. We have attached the STROBE Checklist and submitted as per the criteria.

• Please specify better the primary outcome: did you assess ICU mortality, hospital mortality, or 28-day mortality? Moreover, I would suggest adding the primary outcome result to the main text as well as in the table.

Response: The primary outcome was ICU Mortality and we have edited the manuscript accordingly.

• Please, it could be interesting and could improve the quality of the manuscript to report 1) at least a score or parameters of illness severity (i.e., P/F ratio, Respiratory Rate, SOFA or SAPS score…); 2) the number of patients who needed mechanical ventilation; 3) the kind of steroid administered; 4) the timing of corticosteroid therapy starting compared to ICU admission.

Response: I agree, this could improve the quality of the study. However, we did not collect data on the different scoring systems. Additionally, we only included the steroid doses in dexamethasone equivalence but not the specific steroids used. 

• Univariate statistic tests used are completely described, but I would suggest adjusting for possible confounders (i.e., age, sex, comorbidities, the severity of illness) for a better evaluation of steroid dosage impact on outcome. Moreover, if data permits, try to perform a survival analysis (and try to construct a Kaplan-Meier curve figure).

Response: Unfortunately, we did not collect survival data and thus it was beyond the scope of this study to do Kaplan Meier and survival analysis. 

• I would suggest reporting outcome measures in the outcome results section, not in the population characteristic section.

Response: We have tried to edit the manuscript accordingly. 

• In the limitation section of the discussion, you stated that missing data may have influenced the results. Please report the number of missing data for each variable. 

Response: Thank you. We have included this in the limitation section. Only five patients were excluded from the study due to incomplete records. 

• I would suggest reporting Table 1, joined with Table 2, and reporting the percentage of subjects in Table 2 columns.

Response: We have retained the tables as it is on the flow of the results. 

Reviewer 2 

Reviewer #2: The study is good but, being a retrospective study, it has its own limitations as has been pointed by the authors. A few questions for the authors-

1. They have mentioned high dose of steroids as any dose more than 6 mg dexamethasone, were patients on any other steroids like prednisone, methylprednisolone? When were the steroids started in these patients, as in within how many days of onset of symptoms?

Response: Yes, other types of steroids were used but in the data collection tool we only included steroid doses which were all converted to dexamethasone into dexamethasone equivalence. 

2. Was there any comparison between the survivors and non survivors in terms of oxygen requirements, ventilators-NIV or invasive?

Response: Thank you for this. This may have been a good comparison to do. All the patients included in the study were on ventilatory support, but we did not include the specific type of support. 

3. Was there any correlation between the d Dimer levels amongst the survivors and non survivor groups?

Response: No data was collected on D-dimer levels or any other coagulation parameters. 

4. How many patients were on Remdesevir considering that these were the more severe cases and was there any correlation on this?

Response: This was beyond the scope of our study and has been included as part of the limitations. 

5. How many of these patients also received LMWH or oral anticoagulants and did this affect the outcome of the study?

Response: We did not include data on the anticoagulation. However, as per the hospital policy and guidelines, all patients admitted to critical care unit are put on prophylactic anticoagulation or treatment doses in case of evidence of thrombosis unless there are contraindications. 

6. How many patients underwent a bronchoscopy and BAL for the diagnosis of pneumonia?

Response: Thank you for this. Unfortunately, this was beyond the scope of our study.

---

## [Decision Letter · Decision Letter 1]

17 Mar 2024

PONE-D-23-24666R1OUTCOMES OF DIFFERENT STEROID DOSING REGIMENS IN CRITICAL COVID-19 PNEUMONIA: A RETROSPECTIVE COHORT STUDYPLOS ONE

Dear Dr. Odhiambo,

Thank you for submitting your manuscript to PLOS ONE. After careful consideration, we feel that it has merit but does not fully meet PLOS ONE’s publication criteria as it currently stands. Therefore, we invite you to submit a revised version of the manuscript that addresses the points raised during the review process.

Please revise.

We look forward to receiving your revised manuscript.

Kind regards,

Academic Editor

PLOS ONE

Journal Requirements:

Reviewers' comments:

Reviewer's Responses to Questions

**Comments to the Author**

1. If the authors have adequately addressed your comments raised in a previous round of review and you feel that this manuscript is now acceptable for publication, you may indicate that here to bypass the “Comments to the Author” section, enter your conflict of interest statement in the “Confidential to Editor” section, and submit your "Accept" recommendation.

Reviewer #1: (No Response)

Reviewer #2: All comments have been addressed

Reviewer #3: (No Response)

2. Is the manuscript technically sound, and do the data support the conclusions?

Reviewer #1: No

Reviewer #2: Yes

Reviewer #3: Partly

3. Has the statistical analysis been performed appropriately and rigorously? 

Reviewer #1: No

Reviewer #2: Yes

Reviewer #3: Yes

4. Have the authors made all data underlying the findings in their manuscript fully available?

Reviewer #1: No

Reviewer #2: Yes

Reviewer #3: Yes

5. Is the manuscript presented in an intelligible fashion and written in standard English?

Reviewer #1: Yes

Reviewer #2: Yes

Reviewer #3: Yes

6. Review Comments to the Author

Reviewer #1: Unfortunately, the authors, despite the changes in the manuscript were not able to address any of the instance I raised.

Reviewer #2: All comments have been addressed to satisfaction and the article can be accepted if found to be suitable for publication by the editor.

Reviewer #3: It is difficult to conclude that high doses of steroids do not influence the prognosis.

Can you compare the data that can assess the severity of the 2 groups? Particularly saturation at admission and during progression.

or add this mention to the limitations of this study.

7. PLOS authors have the option to publish the peer review history of their article (what does this mean?). If published, this will include your full peer review and any attached files.

Reviewer #1: No

Reviewer #2: No

Reviewer #3: **Yes: **Abdelbassat Ketfi

---

## [Author Response · Author response to Decision Letter 1]

8 Apr 2024

Response to Reviewer’s Comments

PLOS ONE

OUTCOMES OF DIFFERENT STEROID DOSING REGIMENS IN CRITICAL COVID-19 PNEUMONIA: A RETROSPECTIVE COHORT STUDY 

Thank you for the comments. Please find below the responses to both reviewers’ comments. 

Reviewer #1: Unfortunately, the authors, despite the changes in the manuscript were not able to address any of the instance I raised.

Response: We appreciate the comments and suggestions from Reviewer 1. Some of the suggestions such as reporting at least a score or parameters of illness severity (i.e., P/F ratio, Respiratory Rate, SOFA or SAPS score…) or performing survival analysis was beyond the scope of the current project as the different data was not collected. Furthermore, we attached the STROBE guidelines to report the observational study as requested. We tried to address as much as possible and edited the manuscript accordingly. 

Reviewer #2: All comments have been addressed to satisfaction and the article can be accepted if found to be suitable for publication by the editor.

Response: We appreciate the comments and suggestions from Reviewer 2.

Reviewer #3: It is difficult to conclude that high doses of steroids do not influence the prognosis. Can you compare the data that can assess the severity of the 2 groups? Particularly saturation at admission and during progression or add this mention to the limitations of this study.

Response: We appreciate the comments from Reviewer 3. We included all the patients with critical COVID-19 and on ventilator support as per the inclusion criteria. However, we did not collect any data on the measures of severity such as oxygen saturations. We have, therefore, added it to our limitations as you have advised.

---

## [Decision Letter · Decision Letter 2]

1 May 2024

PONE-D-23-24666R2OUTCOMES OF DIFFERENT STEROID DOSING REGIMENS IN CRITICAL COVID-19 PNEUMONIA: A RETROSPECTIVE COHORT STUDYPLOS ONE

Dear Dr. Odhiambo,

Thank you for submitting your manuscript to PLOS ONE. After careful consideration, we feel that it has merit but does not fully meet PLOS ONE’s publication criteria as it currently stands. Therefore, we invite you to submit a revised version of the manuscript that addresses the points raised during the review process.

Please revise.

We look forward to receiving your revised manuscript.

Kind regards,

Academic Editor

PLOS ONE

Reviewers' comments:

Reviewer's Responses to Questions

**Comments to the Author**

1. If the authors have adequately addressed your comments raised in a previous round of review and you feel that this manuscript is now acceptable for publication, you may indicate that here to bypass the “Comments to the Author” section, enter your conflict of interest statement in the “Confidential to Editor” section, and submit your "Accept" recommendation.

Reviewer #1: (No Response)

Reviewer #2: All comments have been addressed

Reviewer #3: All comments have been addressed

Reviewer #4: All comments have been addressed

2. Is the manuscript technically sound, and do the data support the conclusions?

Reviewer #1: Partly

Reviewer #2: Yes

Reviewer #3: Yes

Reviewer #4: No

3. Has the statistical analysis been performed appropriately and rigorously? 

Reviewer #1: No

Reviewer #2: Yes

Reviewer #3: Yes

Reviewer #4: Yes

4. Have the authors made all data underlying the findings in their manuscript fully available?

Reviewer #1: Yes

Reviewer #2: Yes

Reviewer #3: Yes

Reviewer #4: No

5. Is the manuscript presented in an intelligible fashion and written in standard English?

Reviewer #1: Yes

Reviewer #2: Yes

Reviewer #3: Yes

Reviewer #4: Yes

6. Review Comments to the Author

Reviewer #1: The choice not to collect further descriptive data from the two cohorts, unfortunately, is not agreeable. The severity data of the pathology wouldn't have had solely a descriptive purpose but could have helped the authors to consider possible confounders in the statistical analysis: such as whether patients who had taken higher doses of steroids were the most severe and thus at higher risk of superinfection

Reviewer #2: All queries have been addressed to satisfaction and may be considered for publication if the article meets the other requirements.

Reviewer #3: (No Response)

Reviewer #4: the points mentioned by the authors in the limitation are very important for the study. because of these deficiencies, it is not suitable for publication. also the discussion section is inadequate

7. PLOS authors have the option to publish the peer review history of their article (what does this mean?). If published, this will include your full peer review and any attached files.

Reviewer #1: No

Reviewer #2: No

Reviewer #3: **Yes: **Abdelbassat Ketfi

Reviewer #4: No

---

## [Author Response · Author response to Decision Letter 2]

14 Jun 2024

If the authors have adequately addressed your comments raised in a previous round of review and you feel that this manuscript is now acceptable for publication, you may indicate that here to bypass the “Comments to the Author” section, enter your conflict of interest statement in the “Confidential to Editor” section, and submit your "Accept" recommendation.

Reviewer #1: (No Response)

Reviewer #2: All comments have been addressed

Reviewer #3: All comments have been addressed

Reviewer #4: All comments have been addressed

Response: Thank you for taking time out of your schedule to review our paper and we are happy to note that all your queries were addressed. 

Reviewer #1: The choice not to collect further descriptive data from the two cohorts, unfortunately, is not agreeable. The severity data of the pathology wouldn't have had solely a descriptive purpose but could have helped the authors to consider possible confounders in the statistical analysis: such as whether patients who had taken higher doses of steroids were the most severe and thus at higher risk of superinfection

Response: Having realised that this would improve the quality of the manuscript as was highlighted by the reviewer, we went back and collected data on markers of severity of the disease for instance CRP and oxygen saturation at admission. Due to medical records being paper documents and also limited data available, we aimed at collecting CRP and oxygen saturation. From furtner analyses, it was evident that oxygen saturation at admission was significantly lower in the non-survivors’ group than the survivors’ group whereas as CRP was significantly lower in the high dose group as compared to the standard dose group. 

Reviewer #2: All queries have been addressed to satisfaction and may be considered for publication if the article meets the other requirements.

Response: Thank you for taking time out of your schedule to review our paper and we are happy to note that all your queries were addressed. 

Reviewer #4: The points mentioned by the authors in the limitation are very important for the study. The discussion section is inadequate

Response: Thank you so much for the point you have raised. We have delt with one of the major limitations which was lack of data on the markers of severity such as the level of the inflammatory markers and oxygen saturations at admission and put up an additional paragraph in the discussion section. We believe that this has improved the quality of the manuscript and hopefully accepted for publication.

---

## [Decision Letter · Decision Letter 3]

3 Jul 2024

OUTCOMES OF DIFFERENT STEROID DOSING REGIMENS IN CRITICAL COVID-19 PNEUMONIA: A RETROSPECTIVE COHORT STUDY

PONE-D-23-24666R3

Dear Dr. Odhiambo,

We’re pleased to inform you that your manuscript has been judged scientifically suitable for publication and will be formally accepted for publication once it meets all outstanding technical requirements.

Kind regards,

Academic Editor

PLOS ONE

Additional Editor Comments (optional):

Reviewers' comments:

Reviewer's Responses to Questions

**Comments to the Author**

1. If the authors have adequately addressed your comments raised in a previous round of review and you feel that this manuscript is now acceptable for publication, you may indicate that here to bypass the “Comments to the Author” section, enter your conflict of interest statement in the “Confidential to Editor” section, and submit your "Accept" recommendation.

Reviewer #2: All comments have been addressed

Reviewer #3: All comments have been addressed

2. Is the manuscript technically sound, and do the data support the conclusions?

Reviewer #2: Yes

Reviewer #3: Yes

3. Has the statistical analysis been performed appropriately and rigorously? 

Reviewer #2: Yes

Reviewer #3: Yes

4. Have the authors made all data underlying the findings in their manuscript fully available?

Reviewer #2: Yes

Reviewer #3: Yes

5. Is the manuscript presented in an intelligible fashion and written in standard English?

Reviewer #2: Yes

Reviewer #3: Yes

6. Review Comments to the Author

Reviewer #2: All comments have been addressed to satisfaction and could be considered for publication if it meets the other requirements and has fulfilled the other criteria.

Reviewer #3: (No Response)

7. PLOS authors have the option to publish the peer review history of their article (what does this mean?). If published, this will include your full peer review and any attached files.

Reviewer #2: No

Reviewer #3: No

---

## [Editor Report · Acceptance letter]

8 Jul 2024

PONE-D-23-24666R3 

PLOS ONE

Dear Dr. Odhiambo, 

I'm pleased to inform you that your manuscript has been deemed suitable for publication in PLOS ONE. Congratulations! Your manuscript is now being handed over to our production team.

Kind regards, 

on behalf of

Dr. Robert Jeenchen Chen 

Academic Editor

PLOS ONE